# TypeT5: Seq2seq Type Inference using Static Analysis

**Jiayi Wei, Greg Durrett, Isil Dillig**
Department of Computer Science
University of Texas at Austin
`{jiayi,gdurrett, isil}@cs.utexas.edu`

## Abstract

There has been growing interest in automatically predicting missing type annotations in programs written in Python and JavaScript. While prior methods have achieved impressive accuracy when predicting the most common types, they often perform poorly on rare or complex types. In this paper, we present a new type inference method that treats type prediction as a code infilling task by leveraging CodeT5, a state-of-the-art seq2seq pre-trained language model for code. Our method uses static analysis to construct dynamic contexts for each code element whose type signature is to be predicted by the model. We also propose an iterative decoding scheme that incorporates previous type predictions in the model's input context, allowing information exchange between related code elements. Our evaluation shows that the proposed approach, TypeT5, not only achieves a higher overall accuracy (particularly on rare and complex types) but also produces more coherent results with fewer type errors—while enabling easy user intervention.

## 1    Introduction

In languages like Python and JavaScript, the lack of a static type system makes it harder to maintain and analyze codebases. To address this issue, *gradual typing* (Siek & Taha, 2007) was proposed to allow type annotations to be incrementally added to untyped codebases, thereby marrying the benefits of static typing with the convenience of easy prototyping. As a result, many mainstream programming languages, including Python and JavaScript, have already adopted this idea, and researchers have also developed learning-based techniques to predict missing type annotations (Raychev et al., 2015; Hellendoorn et al., 2018; Wei et al., 2020; Pradel et al., 2020; Allamanis et al., 2020; Pandi et al., 2020; Jesse et al., 2021; Mir et al., 2022; Jesse et al., 2022; Peng et al., 2022).

Meanwhile, with the advent of large-scale pretraining and the explosion of transformer architectures, seq2seq models have proven to be very effective for programming tasks like code comments generation (Panthaplackel et al., 2020), completion (Wang et al., 2021; Ahmad et al., 2021), and synthesis (Li et al., 2022). One particularly attractive feature of such models is that, due to the use of subword tokenization (Gage, 1994; Schuster & Nakajima, 2012; Sennrich et al., 2016), they can generate arbitrary code expressions—including novel identifier names and AST structures—at test time. However, unlike code completion tasks that can often work well with just the surrounding code as context, effective type inference generally requires non-local information, including code fragments that may belong to an entirely different file. For instance, consider a function $f$ that passes a generically named parameter $x$ directly into another function $g$. It can be hard to figure out the type of $x$ by just looking at $f$'s body. When programmers find themselves in such a situation, they often inspect the callers and callees of $f$, sometimes even transitively, in order to figure out the intended type of $x$. Thus, in many cases, looking at the immediate context of a given variable may be insufficient for accurately predicting its type.

Our approach, TypeT5, solves this challenge by using static analysis to identify which parts of the codebase are useful for each prediction. In particular, we construct a so-called *usage graph*, where nodes correspond to code elements (i.e., functions or variables whose types we want to predict) and edges denote a potential user-usee relation between them. Given such a graph, we then encode the users and usees of a given code element in a form that resembles normal code and feeds them as

additional contexts to the transformer model. To take full advantage of the seq2seq paradigm, we also propose an iterative decoding scheme that pass in previous type predictions using the contexts, allowing information to be propagated between distant code elements across the entire codebase.

We have implemented TypeT5 on top of the popular CodeT5 model and use it to synthesize type annotations for untyped Python code. Our evaluation compares TypeT5 with three state-of-the-art type inference tools (Allamanis et al., 2020; Mir et al., 2022; Peng et al., 2022) and a CodeT5 baseline that does not leverage static analysis. The results show that TypeT5 outperforms all baselines by a large margin, while drastically improving the accuracy on rare and complex types. Our ablation studies confirm the benefits of the various modifications we made to the CodeT5 baseline, while an additional type checking experiment shows that the proposed iterative decoding scheme also improves the coherence of the produced type assignments, resulting in fewer type constraint violations. Finally, we explore an alternative use case of our model, where the user interactively inspects the model's predictions and makes necessary corrections. The result demonstrates the usefulness of our approach as a developer tool to annotate entirely untyped projects—on average, the user only needs to to correct one in every five model predictions.

To summarize, this papers makes the following contributions:

- We apply CodeT5 to infer Python type annotations and show significant improvement over prior approaches. To our knowledge, this is the first ML-based technique capable of predicting both parametric and user-defined types.

- We improve the vanilla CodeT5 model by applying static analysis techniques to help the model reason about information beyond local contexts, further boosting its performance.

- We propose an iterative decoding scheme that particularly helps with *coherence*, as measured by the number of type errors reported by the type checker. We additionally propose the novel setting that combines the seq2seq decoding scheme with user intervention.

## 2 OVERVIEW

In this section, we motivate the design of TypeT5 using the example shown in Figure 1. This example features a method `predict` and two functions `eval_on_dataset` and `chuck_srcs`, each of which is implemented in a different file. Given an untyped version of this code, our goal is to automatically infer the type annotations (highlighted in green). This example is challenging for existing type inference techniques due to the heavy use of user-defined types (such as `ChunkedDataset`, `PythonType`, and `ModelWrapper`) and complex parametric type like `dict[int,list[PythonType]]`.

```
🐍 model.py
1    class ModelWrapper:
2        ... # other methods and attributes omitted
3
4        def predict(
5            self,
6            data: ChunkedDataset,
7            n_seqs: Optional[int] = None,
8        ) -> dict[int, list[PythonType]]:
9            pred_types = dict()
10           for batch in data.data:
11               batch["input_ids"] = batch["input_ids"].to(device)
12               preds, _ = self.predict_on_batch(batch, n_seqs)
13               for i, c_id in enumerate(batch["chunk_id"]):
14                   if n_seqs is None:
15                       pred_types[c_id] = preds[i]
16                   else:
17                       span = i * n_seqs : (i + 1) * n_seqs
18                       pred_types[c_id] = preds[span]
19           return pred_types
```

```
🐍 eval.py
1    def eval_on_dataset(
2        model: ModelWrapper,
3        data: TokenizedSrcSet,
4        window_size: Optional[int] = None,
5    ) -> dict[int, list[PythonType]]:
6        window = copy(model.DefaultWindow)
7        if scale_window is not None:
8            window.left_tokens = window_size
9            window.right_tokens = window_size
10
11       chunks = chunk_srcs(data, window)
12       return model.predict(chunks, n_seqs=None)
```

```
🐍 data.py
1    def chunk_srcs(
2        data: TokenizedSrcSet,
3        window: WindowArgs
4    ) -> ChunkedDataset:
5        ... # implementation omitted
6        return ChunkedDataset(...)
```

Figure 1: Simplified code snippets taken from our own codebase. The `eval_on_dataset` function first calls the `chunk_srcs` function to convert the given textual data into equally sized chunks, and it then feed them into the `ModelWrapper.predict` method.

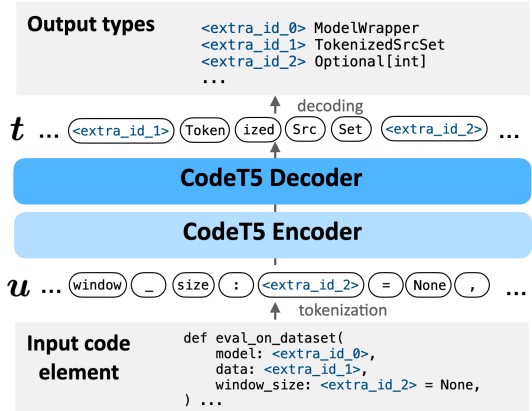

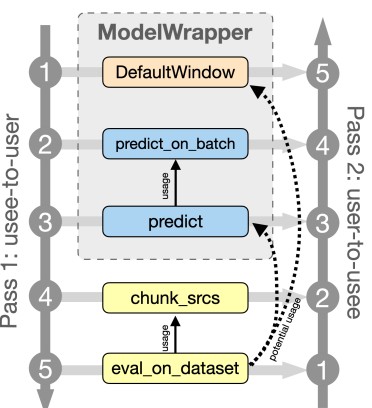

Figure 2: How CodeT5 encodes and decodes source code using BPE. Marker tokens (highlighted in blue) indicate gaps (input) and their corresponding fillers (output).

Figure 3: The two-pass iterative decoding process.

**Type inference as code infilling** In this work, we advocate a new approach that views type inference as an instance of code infilling. Because type annotations can be viewed as missing code fragments, we fine-tune a state-of-the-art code infilling model, namely CodeT5, as our starting point. Since CodeT5 produces sequences of subword tokens using Byte Pair Encoding (Radford et al.; Gage, 1994) (see Figure 2), it can, in principle, predict arbitrary code snippets to fill masked gaps—including type annotations with complex parametric types and user-defined classes.

**Incorporating context through static analysis** By using surrounding code as the prediction context, our fine-tuned version of CodeT5 can relatively easily predict the correct type annotations of some of the variables. For example, based on the names and default values of n_seqs (model.py, line 7) or window_size (eval.py, line 4), CodeT5 can figure out the correct types of these parameters. However, for other parameters such as model in line 2 of eval.py, the surrounding context does not contain enough information to make a reasonable prediction. To see why, observe that ModelWrapper is a new class defined in a separate file, so, (1) it has never been seen during training, and (2) its definition is not available as part of the context. Similarly, it is also very difficult to predict the return type of eval_on_dataset since it directly returns the result of model.predict, whose definition is also not available to the model.

To address this issue, our approach enhances the prediction context of CodeT5 using static analysis. The details of how we construct the context from static analysis will be described in subsection 3.3, but, in a nutshell, our approach analyzes the user-usee relations among code elements and pulls in the relevant definitions into the context. For example, when the model is making a prediction for eval_on_dataset, the context includes the definitions of DefaultWindow and predict, which are defined in ModelWrapper and invoked at line 6 and line 12 of eval.py, respectively.

**TypeT5 architecture** As there are many dependencies between different code elements, making *independent* type predictions for each code element is not ideal. For example, to infer the return type of eval_on_dataset, we would need to know the return type of ModelWrapper.predict, which depends on the return type of the self.predict_on_batch (model.py, line 12). However, since there is limited context that can be fed to the model, it is not feasible to include all transitive users and usees. To deal with this problem, TypeT5 utilizes an iterative decoding scheme that allows conditioning on previous type predictions. In more detail, our decoding scheme first sorts the user-usee graph topologically and then performs two sequential prediction passes, first from usee to users and then going in the reverse direction, as illustrated in Figure 3. To see why both of these directions are useful, observe that the return type of eval_on_dataset depends on predict, which in turn depends on predict_on_batch. Thus, propagating information from callees to callers is clearly useful. Conversely, consider predicting the type of data, the first parameter of predict. Since the return value of chunk_srcs is passed as the first argument of predict, propagating information in the reverse direction can also be helpful.

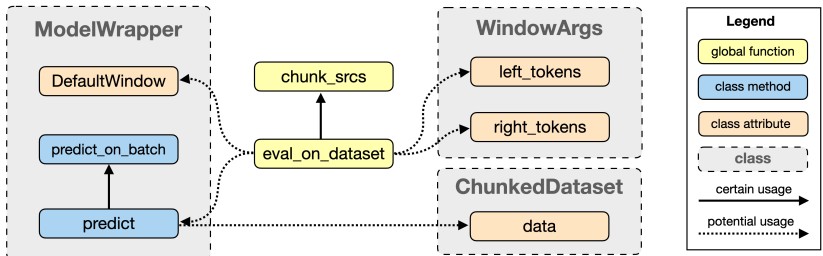

Figure 4: The usage graph corresponding to the code snippets in Figure 1. `WindowArgs` and `ChunkedDataset` are assumed to be defined elsewhere in the same project.

## 3 METHODS

### 3.1 USING CODET5 FOR TYPE PREDICTION

We formulate type prediction as a sequence-to-sequence (seq2seq) task. Let $\boldsymbol{u} = (u_1, \ldots, u_n)$ represent a sequence of code tokens, where each token is a single untyped code element $e$ (function or variable). We insert into $\boldsymbol{u}$ indexed marker tokens (`<extra_id_i>`) at each point where we wish to predict types and let the model predict $\boldsymbol{t} = (t_1, \ldots, t_m)$, the token sequence that encodes types for the marked locations in $\boldsymbol{u}$. Note that $\boldsymbol{t}$ only contains the types, no other code tokens, in the format `<extra_id_1> [type 1 tokens] <extra_id_2> [type 2 tokens]`, etc. We use the same tokenizer as CodeT5, which allows encoding any Python expression as a (typically short) sequence of subword tokens.

Our baseline CodeT5 model is a Transformer seq2seq model $P(\boldsymbol{t} \mid \boldsymbol{u}, \bar{\boldsymbol{u}})$ placing a distribution over type sequences $\boldsymbol{t}$ given the raw code sequence $\boldsymbol{u}$ and its surrounding code $\bar{\boldsymbol{u}}$. This process is shown in Figure 2, and $\bar{\boldsymbol{u}}$ is omitted for clarity.

Our improved model, TypeT5, replaces the surrounding code $\bar{\boldsymbol{u}}$ with a context constructed from static analysis, which we denote as $\boldsymbol{s}$. Thus, we can write the model as $P(\boldsymbol{t} \mid \boldsymbol{u}, \boldsymbol{s})$. Note that, in both models, we remove all comments and Python docstrings as a pre-processing step, as was done in prior work. We now show how $\boldsymbol{s}$ is constructed from static analysis.

### 3.2 BUILDING THE USAGE GRAPH

To overcome the limitations of using *only* the surrounding code as context, our approach relies on static analysis to extract relevant global information about the codebase. In particular, our analysis constructs a *usage graph*, whose nodes correspond to code elements and edges represent a direct usage. For example, if $x$ and $y$ are both functions, an edge from $x$ to $y$ means that $x$ calls $y$. Similarly, for variables, an edge from $x$ to $y$ means that the value of $x$ depends on $y$.

The usage graph for our previous example is shown in Figure 4. We show two types of usages: a certain usage, shown as solid arrows, and a potential usage, shown as dotted arrows. A certain usage is one that can be statically determined without knowing the types of the involved variables. For example, the global function `chunk_srcs` is directly used by `eval_on_dataset`, and we also know that `predict_on_batch` is called by `predict` (`self` method call on line 12). For other cases like a non-`self` attribute access or method call, the target depends on the type of the receiver, so we first collect all attributes and methods with a matching name from the current project and then generate a potential usage for each of them. We give more details of usage graph construction in subsection A.1.

### 3.3 CONSTRUCTING MODEL INPUTS

TypeT5 leverages the usage graph $\mathcal{G}$ to construct the inputs to our model. In particular, we define the model input for a code element $e$ to be a concatenation of four parts: the preamble $\boldsymbol{s}_{\text{pre}}$, usees $\boldsymbol{s}_{\text{usee}}$, main code $\boldsymbol{u}$, and users $\boldsymbol{s}_{\text{user}}$, which are constructed as follows (see subsection A.9 for a concrete example).

- **Preamble**: The main purpose of $s_{\text{pre}}$ is to help the model map each local name to the definition that it is referring to, so the preamble includes all import statements. Additionally, to help the model see which types are available in the current scope, the preamble also includes the headers of all class definitions as well as all the type variable declarations from the current file.

- **Main code**: For the code element $e$ of interest, we use its definition to construct $\boldsymbol{u}$. If $e$ is a function, $\boldsymbol{u}$ is just its source code, and if $e$ is a variable, $\boldsymbol{u}$ includes all top-level assignment statements in which $e$ appears as the left hand side. Additionally, if $e$ is a method or attribute of a class $C$, we also indent $\boldsymbol{u}$ and prepend it with the header of $C$, as shown in Figure 7.

- **Users**: Given the usage graph $\mathcal{G}$, we construct $s_{\text{user}}$ by including the source tokens of all elements from $\text{users}(\mathcal{G}, e)$. Since these elements can come from different source files, we also prepend each element with a special comment denoting the Python module it comes from. Note that these elements can optionally contain the types predicted by our TypeT5 model, as described later in subsection 3.4.

- **Usees**: $s_{\text{usee}}$ contains not just the direct users of $e$, but also anything that is used in the user context $s_{\text{user}}$. i.e., $s_{\text{usee}}$ contains the elements from $\text{usees}(\mathcal{G}, e) \cup \bigcup_{e' \in \text{users}(\mathcal{G}, e)} \text{usees}(\mathcal{G}, e')$. Since this generally contains many more elements than $s_{\text{user}}$, we only use the (typed or untyped) *signatures* of the elements to construct $s_{\text{usee}}$.

We limit the total input size to be at most 4096 subword tokens and cut off any exceeding tokens from both left and right, centered around the main code. Context elements involving certain usages are arranged to be closer to the center so that they are always prioritized over potential usages.

## 3.4 Iterative Decoding Inference

We now describe how to conduct inference in our base model as well as our context-augmented model using an iterative decoding scheme.

**CodeT5 decoding:** Given our trained model $P(\boldsymbol{t} \mid \boldsymbol{u}, \bar{\boldsymbol{u}})$, we can infer a most likely set of types $\boldsymbol{t}$ for $\boldsymbol{u}$ (with surrounding context $\bar{\boldsymbol{u}}$) using beam search. Our implementation performs joint prediction of the output types for a single code block $\boldsymbol{u}$, since later types in the block condition on the predictions of earlier types. However, note that both $\boldsymbol{u}$ and $\bar{\boldsymbol{u}}$ are always *completely untyped* code: while we condition on previous types as we predict, these are not inserted into the prediction context for the next element.[1]

**TypeT5 iterative decoding:** Part of our motivation for including the context $s$ is to exploit its type information at inference time. Crucially, this requires $s$ to be *typed*. However, the contexts that are drawn from the original codebase are not typed, so TypeT5 iteratively adds type signatures to these contexts using its own predictions. Let $\mathcal{M}$ be the type assignment produced by the model, which maps each code element $e$ to its predicted type $\boldsymbol{t}_e$, and denote $\mathcal{M}(s)$ as the context obtained by annotating the elements in $s$ according to $\mathcal{M}$. Starting with an empty $\mathcal{M}$ (which leaves any context $s$ unchanged), TypeT5 then iteratively updates $\mathcal{M}$ using an iterative decoding scheme that traverses the usage graph $\mathcal{G}$ twice, as shown in Figure 3. The first prediction pass follows the usee-to-user order,[2] while the second pass goes in the reverse direction to allow for bidirectional information flow. At any given time step, we can denote the model's prediction for element $e$ as drawn from $P(\boldsymbol{t}_e \mid \boldsymbol{u}_e, \mathcal{M}(s_e))$, and the predicted types $\boldsymbol{t}'_e$ are then used to update $\mathcal{M}$ such that $\mathcal{M}(e) = \boldsymbol{t}'_e$.

## 3.5 Training

To save computation and improve parallelism during training, we use the available human annotations as a gold type assignment $\mathcal{M}^*$ instead of letting the model condition on its own prediction. Note that this type assignment is generally incomplete and may not contain a label for every missing type. We train the model to maximize the log likelihood of predicting the ground truth, i.e., $\log P(\boldsymbol{t}^*_e \mid \boldsymbol{u}_e, \mathcal{M}^*(s_e))$, for every element $e$ where $\boldsymbol{t}^*_e$ is available, using teacher-forcing. We train the CodeT5 baseline model similarly on the same dataset.[3]

---

[1] In our experiments, including predicted types actually hurts the performance due to exposure bias.

[2] This requires performing a topological sort over $\mathcal{G}$. When $\mathcal{G}$ is not a DAG, we break the cycles arbitrarily.

[3] Note that without this training (fine-tuning) step, the original CodeT5 model performs very poorly as it tends to predict arbitrary Python expressions that are not types.

Table 1: Basic statistics of our two datasets.

| | BetterTypes4Py | | | InferTypes4Py |
| | train | valid | test | test |
|---|---|---|---|---|
| Projects | 573 | 40 | 50 | 3 |
| Nonempty files | 16.5K | 1098 | 949 | 99 |
| Lines of code | 2.4M | 174K | 139K | 21K |
| Top-level type slots | 541K | 38.2K | 28.4K | 4.6K |
| Top-level user-added types | 275K | 19.3K | 15.8K | 2.7k |
| Rare type ratio | 25.7% | 23.3% | 35.0% | 33.8% |
| Complex type ratio | 20.4% | 16.6% | 20.8% | 33.4% |
| Average type size | 1.42 | 1.33 | 1.43 | 1.72 |

## 4 EXPERIMENTS

We implement TypeT5 in Python, whose the source code and model weights can be found on GitHub[4]. We list the hyperparameters and hardware details in subsection A.6. Below, we first compare TypeT5 against three state-of-the-art type inference systems for Python, namely Typilus (Allamanis et al., 2020), Type4Py (Mir et al., 2022), and HiTyper (Peng et al., 2022). We then present ablation studies to evaluate different factors contributing to the model's performance.

### 4.1 EVALUATION SETUP

**Datasets** In our evaluation, we predict the type annotations for *top-level* code elements of each Python project. These include all class attributes, methods, top-level functions, and global variables. We treat any existing user-added type annotations as the ground truth, and we use per type accuracy as the main performance metric. Following a setting similar to that of Allamanis et al. (2020) and Wei et al. (2020), we split our dataset per Python project. This way, types newly defined in the test projects will not have been seen during training. Such a setting is more challenging than splitting the dataset per file, as is done in Type4Py and other work (Pradel et al., 2020; Mir et al., 2022), but more closely reflects the model's real-world performance (Gruner et al., 2022).

Our main dataset, **BetterTypes4Py**, is constructed by selecting a high-quality subset from the Many-Types4Py dataset (Mir et al., 2021), which was used to train Type4Py. We describe the selection criteria in subsection A.2. Since our model is fine-tuned from the CodeT5 model (which may have already been pre-trained on some of the test repositories in the aforementioned dataset), we additionally construct **InferTypes4Py**, a test set derived from the source code of Typilus, Type4Py, and our own tool, none of which were used as CodeT5's (pre-)training data. We further discuss the potential code duplication issue (subsection A.3) and label quality issue (subsection A.4) in the appendix.

We summarize key properties of both datasets in Table 1. We define the size of a type as the number of type constructors in its body[5] and categorize a type as **simple** if its size is 1, and **complex** otherwise. We also categorize a type as **rare** or **common** depending on whether it contains a rare type constructor that is not from the top-100 most frequent type constructors in our training set.

**Accuracy metrics.** Since Python allows the same type to be written in different syntactic forms,[6] we first perform type normalization to convert both the predicted and ground-truth types into a canonical form. The details of this normalization step can be found in subsection A.5, and we use the term **full accuracy** to refer to the accuracy against all human annotations after normalization. To better compare with prior work, we also define **adjusted accuracy** (our main metric), which (1) filters out all `None` and `Any` labels (as in prior work); (2) converts fully qualified names to simple names (e.g., `Tensor` instead of `torch.Tensor`) since some prior approach does not output correctly qualified types; (3) rewrites any outermost `Optional[T]` and `Final[T]` into `T` since they

---

[4]Available at `https://github.com/utopia-group/TypeT5`.

[5]e.g., both `int` and `foo.Bar` has a size of 1, whereas `dict[str, foo.Bar]` has a size of 3.

[6]e.g., both `Union[int,None]` and `Optional[int]` refer to an integer that can also be `None`, and both `list` and `List[Any]` refer to a python list with untyped elements

Table 2: Accuracy comparison on **common** types.

| | BetterTypes4Py | | | | | InferTypes4Py | | | | |
| | **full** | | **adjusted** | | **base** | **full** | | **adjusted** | | **base** |
| | all | all | simple | complex | all | all | all | simple | complex | all |
|---|---|---|---|---|---|---|---|---|---|---|
| Typilus | n/a | 54.05 | 55.12 | 33.23 | 60.37 | n/a | 52.33 | 52.19 | 53.91 | 64.67 |
| Type4Py | n/a | 50.34 | 51.91 | 32.14 | 47.51 | n/a | 32.08 | 33.47 | 16.54 | 29.83 |
| HiTyper | 59.20 | 54.28 | 57.70 | 26.44 | 59.01 | 45.67 | 43.54 | 46.00 | 19.27 | 47.99 |
| CodeT5 | 76.74 | 78.04 | 82.43 | 53.03 | 82.44 | 77.83 | 78.06 | 85.31 | 63.41 | 81.87 |
| TypeT5 | **79.24** | **81.43** | **85.69** | **56.75** | **84.82** | **81.75** | **82.95** | **87.62** | **72.78** | **84.17** |

Table 3: Accuracy comparison on **rare** types.

| | BetterTypes4Py | | | | | InferTypes4Py | | | | |
| | **full** | | **adjusted** | | **base** | **full** | | **adjusted** | | **base** |
| | all | all | simple | complex | all | all | all | simple | complex | all |
|---|---|---|---|---|---|---|---|---|---|---|
| Type4Py | n/a | 12.37 | 13.17 | 4.05 | 14.15 | n/a | 0.25 | 0.14 | 0.98 | 0.17 |
| HiTyper | 10.30 | 25.51 | 27.59 | 9.79 | 29.33 | 9.36 | 9.36 | 10.79 | 1.19 | 12.33 |
| CodeT5 | 49.47 | 52.95 | 57.28 | 34.26 | 57.65 | 51.64 | 53.28 | 59.97 | 30.62 | 66.47 |
| TypeT5 | **58.56** | **61.47** | **65.21** | **40.22** | **68.44** | **53.44** | **56.27** | **61.50** | **36.92** | **69.23** |

tend not to be used consistently across programmers.[7] Finally, we also define a **base accuracy** metric that is the same as adjusted accuracy except that it only checks the outermost type (e.g., `Dict[str,List]` will match any `Dict` but, for example, not `Mapping`.)

## 4.2 COMPARING TYPET5 WITH OTHER APPROACHES

We compare TypeT5 with the basic CodeT5 model described in subsection 3.4 as well as the released versions of three other state-of-the-art approaches from prior work.[8] Typilus (Allamanis et al., 2020) models the program as a graph and applies a graph neural network to compute the types of variables. The original Typilus model can predict from a set of common types as well as (nonparametric) user-defined types, but their released model can only predict common types, so we only evaluate its performance on common types. Type4Py (Mir et al., 2022) uses variable names and the surrounding code to compute an embedding and performs type prediction via a nearest-neighbor search in the type embedding space. We run the released Type4Py model via its web interface. HiTyper (Peng et al., 2022) combines the strengths of a rule-based type inference algorithm with ML type inference models by only invoking the ML model on places where the inference algorithm gets stuck and deducing the types elsewhere using typing constraints. Its implementation uses Type4Py as the default ML backend, which we use to run HiTyper.

We show each tool's accuracy in Table 2 (on common types) and Table 3 (on rare types), and we put the combined results in appendix (Table 6). Since HiTyper was only able to make a predition for about 67% of all labels, we report its performance on this subset. From Table 2, we can make the following observations. (1) Our CodeT5 baseline model already outperforms all prior approaches by a large margin, demonstrating the advantage of using a seq2seq pre-trained language model. (2) TypeT5 further improves the adjusted accuracy by 3.4% and 4.9% on the two datasets. (3) Type4Py's performance on InferTypes4Py dataset is significantly lower than on BetterTypes4Py, likely because Type4Py was originally trained on some of the test files in BetterTypes4Py.[9] Looking at Table 6, we see that both CodeT5 and TypeT5 dramatically outperform the other approaches on rare types. Moreover, TypeT5 achieves the largest improvements on types that are both rare and complex, improving upon CodeT5 by about 6% on both datasets, suggesting that that global information is especially important in these cases. For a qualitative analysis, we also show TypeT5's outputs on a real-world example in subsection A.9.

---

[7]e.g., the type checker `mypy` has an option to enable implicit `Optional` types, so it would not be possible for the model to know if it should output `Optional[T]` or `T` just from the untyped code.

[8]Since our approach benefits from CodeT5's large-scale pre-training across 8 different programming languages, we use a smaller training set than prior work and do not retrain these prior approaches on our dataset.

[9]The accuracies reported in the original Type4Py paper are much higher than we measured here. We analyze this discrepency in subsection A.7.

Table 4: Performance of different model modifications. All models are retrained with the corresponding inputs.

| Modification | Accuracy | Type Error |
|---|---|---|
| No Preamble | 64.20 | 6067 |
| No Users | 71.20 | 7053 |
| No Usees | 67.25 | 7332 |
| Nonincremental | 72.52 | 5720 |
| Original (TypeT5) | **73.02** | **5087** |

Table 5: Performance of different decoding strategies. The same TypeT5 model weights are used for different decoding strategies.

| Strategy | Accuracy | Type Error |
|---|---|---|
| Independent | 71.68 | 6876 |
| Random | 71.66 | 6215 |
| UserToUsee | 70.67 | 7415 |
| UseeToUser | 72.65 | 6402 |
| TwoPass (TypeT5) | **73.02** | **5087** |

## 4.3 ABLATIONS ON TYPET5

We next present a series of ablations that evaluate how various factors contribute to TypeT5's performance. In addition to accuracy (on all types), we also report the **type error count** as a way to estimate how *coherent* the model's predictions are.[10] Note that we only report errors that are directly related to type coherence (since the type checker also reports errors unrelated to type coherence, such as unresolved import statements. We give more details about this metric in subsection A.8.).

**How do different components contribute to the model's performance?** To evaluate the impact of each context element, we remove one component at a time and *retrain the model* accordingly. In particular, the **No Preamble**, **No Users**, and **No Usees** ablations correspond to removing the $s_{\mathrm{pre}}$, $s_{\mathrm{user}}$, and $s_{\mathrm{usee}}$ context elements (introduced in subsection 3.3) from the model's input, respectively. The **Nonincremental** model does not perform iterative decoding and is trained to condition on an untyped context. We use the same input size limit (4096 subword tokens) for all models, so a model that does not utilize one kind of information have more space for other kinds. We show both the adjusted accuracy on all types and type error count in Table 4 (and show other accuracy metrics in Table 7). We can see that (1) all components improve the overall accuracy, with the preamble having the largest impact, and (2) while the iterative decoding scheme only improves overall accuracy slightly, it significantly improves the model's coherence, resulting in 12% fewer type errors.

**How do different decoding strategies compare?** In addition to the two-pass iterative decoding strategy introduced in subsection 3.4, we also test four other decoding strategies: (1) **Independent**, which independently predicts the type signature for each element without conditioning on the model's own prediction (same as the Nonincremental model except not retrained); (2) **Random**, which visits each element once following a random order; (3) **UserToUsee**, which visits the users before the usees; (4) **UseeToUser**, which visits the usees before the users. The results are shown in Table 5 (and show other accuracy metrics in Table 8). We can see that our proposed TwoPass decoding scheme yields the largest accuracy and type error improvement over Independent; whereas UserToUsee performs worse than Independent, suggesting that bad decoding ordering can have adverse effects on the model's performance.

## 4.4 USER-GUIDED INTERACTIVE DECODING

Compared to prior work, our approach has a unique strength: because the model can condition on previous types, it allows easy user intervention by conditioning on any corrections made by the user. We thus explore an alternative use case of our model where the user interactively inspects each type signature predicted by the model and makes any necessary corrections before the model moves on to the next prediction. We emulate this interactive process using the ground-truth human annotations $\mathcal{M}^*$, and we modify the usee-to-user decoding process to let the model predict the types of each element $e$ (as before), but then override the predicted types $t_e$ with the corresponding human annotations $\mathcal{M}^*(e)$ if $e \in \mathcal{M}^*$. On the BetterTypes4Py dataset, this interactive decoding process achieves a full accuracy of **78.04%** and an adjusted accuracy of **79.37%**—meaning that

---

[10]Accuracy does not always correlate with coherency. e.g., when we have `x = y`, and `x` and `y` are equally likely to be a `str` or an `int`, a coherent type assignment needs to ensure that `x` and `y` always have the same type, even if this requirement does not lead to a higher accuracy in expectation.

on average, it only requires the user to correct one in every five model-predicted types to fully annotate an entirely untyped project from scratch. Note that only 59% of the test set contains a user type annotation, so some incorrect type annotations may not be corrected immediately and can get propagated to other places. Thus, in an interactive real-world setting, we can expect the actual accuracy to be even higher.

## 5 RELATED WORK

**Deep Learning Type Inference**    Most prior approaches predict user-defined types via some form of type embedding matching. For example, LambdaNet (Wei et al., 2020) tackles this challenge by combining graph neural networks with a pointer network. Typilus (Allamanis et al., 2020) performs nearest-neighbor search in the type embedding space. However, neither approach can handle the unbounded type space induced by parametric types. TypeBert (Jesse et al., 2021) is the first type inference method based on pre-trained transformer models and has demonstrated superior performance in predicting common types. Jesse et al. (2022) later improved TypeBert using deep similarity learning to better support user-defined types. Different from our work, TypeBert does not use a seq2seq model or construct context from static analaysis. Apart from just using the source code, other forms of information have also been utilized for type prediction. Both TypeWriter (Pradel et al., 2020) and HiTyper (Peng et al., 2022) combines type checking with inference-time search to avoid generating type errors. OptTyper (Pandi et al., 2020) explicitly models type constraints by turning them into a continuous optimization objective. NL2Type (Malik et al., 2019) focuses on predicting types from natural language information such as code comments and documentation.

**Retrieval-based models in NLP**    Similar to our context augmentation with static analysis, a number of methods have been developed to augment pre-trained models with context in natural language processing (NLP). For open-domain question answering (Chen et al., 2017), approaches like REALM Guu et al. (2020) and DPR (Karpukhin et al., 2020) can retrieve knowledge relevant to a query, and retrieval-augmented generation (Lewis et al., 2020) and its extensions (e.g., Fusion-in-Decoder (Izacard & Grave, 2020)) have shown that it is possible to generate longer outputs using this information. RETRO (Borgeaud et al., 2021) and WebGPT (Nakano et al., 2021) take this to a web-scale extreme. However, we are also able to leverage static analysis based on the usage graph, which has no analogue for text.

**Linking with T5**    While our use of CodeT5 for type prediction is novel to our knowledge, T5 (Raffel et al., 2019) has been applied to a range of NLP tasks like summarization. Most similar to ours is its use for entity linking (Petroni et al., 2021): Systems in this vein generate names of entities token by token (De Cao et al., 2020) and are able to generalize to new entities like TypeT5. However, the presence of ad hoc types for each new context and the types of context clues needed are very different in the code setting than in natural language.

**Structured Prediction**    Our iterative decoding process can be viewed as applying a learned policy to perform structured prediction (BakIr et al., 2007; Daumé et al., 2009). In this work, our training scheme can be viewed as behavior cloning since the model directly conditions on ground-truth human annotations, which can lead to distributional mismatch between training and inference time. Applying more advanced training schemes such as Scheduled Sampling (Bengio et al., 2015), DAgger (Ross et al., 2011), or reinforcement learning (Sutton & Barto, 2018) may help further boost the performance of iterative decoding.

## 6 CONCLUSION

In this work, we showed that TypeT5, a system integrating CodeT5 with lightweight static analysis and a new decoding scheme, is effective at predicting types for untyped codebases. We show that fine-tuning a pre-trained code model already results in a strong baseline, outperforming systems from past work, and that incorporating the right context with static analysis is particularly helpful to enable accurate prediction on more challenging cases. We believe that even larger models can further improve the accuracy, but they will still need to leverage the methods in this work to achieve high accuracy on rare and complex types.

ACKNOWLEDGMENT

This work is partly funded by NSF Award CCF-1918889. We would like to express our sincere gratitude to Miltiadis Allamanis and Amir Mir for generously helping us in setting up their tools and completing the comparison experiments. We would also like to thank members of the UTOPIA group, especially Jocelyn Chen, for their assistance in managing the GPU server for the experiments.

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

# A APPENDIX

## A.1 CONSTRUCTING THE USAGE GRAPH

To resolve the user-usee relations at the project level (used in subsection 3.2), we built a custom static analysis pipeline on top of the `libcst` library. We use `libcst` to parse Python files and also rely on its utilities to resolve symbol references within the same file (i.e., it tells us whether a local name refers to a function defined in the current file or comes from an import statement, etc.) We then implement custom import resolution logic (following Python's module rules) and combine it with `libcst` to resolve all certain usages within the current project. For unresolved usages with a syntactic form matching a class attribute/method usage, we generate a potential usage to each class members with a matching name. Since the involved static analysis operations are fairly lightweight and we parallelize the construction of different graphs on CPUs, the time spent on constructing the usage graphs only makes up a tiny fraction of the total training or inference time. For example, using 28 Python processes, it only takes about 8 minutes to process the entire BetterTypes4Py training set, including the time spent on other tasks such as parsing, code transformation, and tokenization.

While our implementation limits the analysis scope to the current project, it is straight-forward to extend the analysis to also include usages involving library definitions, which may help the model see type signatures of uncommon library APIs. We did not do this in our experiments mainly due to the manual effort it would require to install the correct library dependencies for all the projects in our datasets.

## A.2 CONSTRUCTING THE BETTERTYPES4PY DATASET

Our dataset is constructed from the ManyTypes4Py dataset as follows: we first filter out the GitHub repositories that are no longer accessible or that fail to download within 10 seconds. This leaves us with 4890 out of the 5996 original projects. Then, we discard projects that have not been updated for more than one year (to avoid outdated library APIs), reducing the number of repositories to 1218. To limit the influence of any particular project, we also filter out the 37 projects with more than 50K lines of code. Finally, to exclude those projects that have very few type annotations, we compare the number of type annotations $n_t$ of each project with its lines of code $n_c$ and filter out those with a $n_t$-to-$n_c$ ratio less than 1:20. This gives us our final set of 663 projects. We then randomly select 50 test and 40 validation projects and use the rest for training.

## A.3 CODE DUPLICATION

Code duplication (Allamanis, 2019) has been shown to have adverse effects on the performance of ML models and can blur the evaluation results. Hence, prior work like Type4Py applies file-level deduplication to remove duplicated source files. However, this is hard to do in our project-based setting since we need all files to be present during inference. To address this, we have manually verified that our InferTypes4Py dataset does not contain files that are copy-pasted from elsewhere. We also run the popular code duplication tool `jscpd`[11] on our test set to detect duplicated code blocks. The analysis shows that there is relatively little duplication in the dataset (only around 4% of duplicated lines), and the majority of these duplications came from the same project rather than across projects, so we believe code duplication is not a major issue under our by-project evaluation.

## A.4 LABEL QUALITY

In our evaluation, developer-provided type annotations are used as the ground truth, which are not always accurate or coherent (Ore et al., 2018). This partially motivated us to construct the Infer-Types4Py dataset, which consists of high-quality type annotations with a relatively low error rate. In particular, our own codebase (which are included in InferTypes4Py) makes heavy use of type annotations throughout the development process and is continuously type-checked by VSCode. As a very rough metric to approximate label quality, we report the coherence error (defined in subsection A.8) of the human labels on both datasets: On BetterTypes4Py and InferTypes4Py, the average coherence error per human annotation is 0.019 and 0.045, respectively.

---

[11]https://github.com/kucherenko/jscpd

Table 6: Accuracy comparison on all types (**common + rare**).

| | BetterTypes4Py | | | | | InferTypes4Py | | | | |
|---|---|---|---|---|---|---|---|---|---|---|
| | **full** | **adjusted** | | | **base** | **full** | **adjusted** | | | **base** |
| | all | all | simple | complex | all | all | all | simple | complex | all |
| Type4Py | n/a | 34.52 | 35.87 | 19.68 | 35.61 | n/a | 21.11 | 22.35 | 9.61 | 22.64 |
| HiTyper | 42.53 | 41.95 | 44.86 | 19.02 | 47.88 | 34.83 | 32.50 | 35.11 | 11.40 | 39.23 |
| CodeT5 | 67.07 | 67.47 | 72.12 | 44.05 | 73.44 | 68.80 | 68.95 | 75.14 | 54.04 | 77.78 |
| TypeT5 | **71.89** | **73.02** | **77.07** | **49.72** | **78.87** | **71.98** | **73.15** | **77.16** | **62.66** | **80.20** |

## A.5 TYPE NORMALIZATION

To compute the accuracy metrics in subsection 4.1, we recursively apply the following steps to normalize a Python type:

1. Rewrite any `Optional[T]` to `Union[T,None]`.
2. Sort the arguments of `Union` types and flatten any nested `Union`s.
   e.g., rewrite `Union[B,Union[C,A]]` into `Union[A,B,C]`.
3. If all type arguments are `Any`, drop them all. e.g., rewrite `List[Any]` to `List`.
4. Capitalize the names of basic types. e.g., rewrite `list` to `List`.

## A.6 HYPERPARAMETERS AND RUNNING TIMES

We initialize our model's weights from the CodeT5 model provided by Huggingface Transformers library (Wolf et al., 2020) and train our model for exactly one epoch using the library's default optimizer configuration (AdamW with a base learning rate of 2e-5 and a weight decay of 0.01.) During inference, we use beam search with a beam width of 16 and diversity penalty of 1.0. We adaptively set the maximal output sequence length to be $16n + 10$, where $n$ is the number of types to be predicted in the input.

We set the size limit of preamble, usee Context, main code, and user Context to 1000, 2048, 512, 1536, respectively, both during training and test time. Note that preamble uses the space within usee context, so the total maximal input size is 4096.

Training the model took about 11 hours on a single Quadro RTX 8000 GPU with 48GB memory. Performing the two-pass inference on the BetterTypes4Py test set takes about 4 hours, whereas performing a single-pass inference (e.g., UseeToUser) takes about half the time. For comparison, training CodeT5 model on the same machine takes about 3.7 hours, and the corresponding evaluation takes about 0.5 hour.

## A.7 WHY TYPE4PY HAS MUCH LOWER PERFORMANCE ON OUR DATASET

We have discussed our experiments with the authors of Type4Py and believe that the discrepancy we observe is likely due to the combination of two reasons: First, while our evaluation only counts the type annotations on all top-level APIs, Mir et al. (2022) includes all local variables in their evaluation as well. Second, while we only evaluate on human annotations, they also include machine-inferred type annotations (via the type checker Pyre) as ground-truth labels. As a result, the distributions of labels reported by the two papers are significantly different: in our setting, function annotations (parameters + return types) constitute the majority (87%) of the labels, whereas in Mir et al. (2022), their portion is significantly smaller, merely 18%. This suggests that their label set is likely inflated by simple labels inferrable from the type checker, which explain the performance drop when evaluated on our datasets.

## A.8 MEASURING TYPE COHERENCE USING TYPE ERRORS

To estimate the type coherence (subsection 4.3), we call the type checker `MyPy`[12] on codebases annotated with the types predicted by the model. Since not all errors reported by `MyPy` are type

---

[12]http://mypy-lang.org/.

Table 7:  Detailed comparison of different model modifications on BetterTypes4Py. All models are retrained with the corresponding input format. Accuracy is measured on all types.

| Modification | full | Accuracy adjusted | | | base | Type Error |
|---|---|---|---|---|---|---|
| | all | all | simple | complex | all | |
| No Preamble | 63.03 | 64.20 | 71.22 | 33.51 | 69.98 | 6067 |
| No Usees | 67.70 | 67.15 | 70.27 | 48.09 | 73.01 | 7053 |
| No Users | 70.22 | 71.20 | **77.87** | 41.34 | 77.26 | 7332 |
| Nonincremental | 71.86 | 72.52 | 77.23 | 47.26 | 78.47 | 5720 |
| Original (TypeT5) | **71.89** | **73.02** | 77.07 | **49.72** | **78.87** | **5087** |

Table 8:  Detailed comparison of different decoding strategies on BetterTypes4Py. The same TypeT5 model weights are used for different decoding strategies. Accuracy is measured on all types, and the UserGuided decoding is introduced in subsection 4.4.

| Modification | full | Accuracy adjusted | | | base | Type Error |
|---|---|---|---|---|---|---|
| | all | all | simple | complex | all | |
| Independent | 70.87 | 71.68 | 75.99 | 46.92 | 77.58 | 6876 |
| Random | 70.68 | 71.66 | 75.68 | 47.89 | 77.65 | 6215 |
| UserToUsee | 69.98 | 70.67 | 74.48 | 47.60 | 76.46 | 7415 |
| UseeToUser | 71.58 | 72.65 | 76.80 | 48.88 | 78.48 | 6402 |
| TwoPass (TypeT5) | **71.89** | **73.02** | **77.07** | **49.72** | **78.87** | **5087** |
| UserGuided | 78.09 | 79.36 | 83.27 | 58.61 | 84.02 | n/a |

errors or are related to type coherence, we only count the errors with the following 5 error codes, whose meaning according to `MyPy`'s documentation are:

- **attr-defined** checks that an attribute is defined in the target class or module when using the dot operator.

- **arg-type** checks that argument types in a call match the declared argument types in the signature of the called function.

- **return-value** checks that the returned value is compatible with the type signature of the function.

- **assignment** checks that the assigned expression is compatible with the assignment target.

- **name-defined** checks that a name is defined in the current scope.

Note that this metric does have its limitations. One undesired property we found is that it can favor predicting a non-existing type over an incorrect type. For instance, when the model predicts an incorrect type on a function, the type checker will check all the usages of that function against this declared type and will thus likely report multiple errors. However, if the model predicts a non-existing type, the type checker will only report a single `name-defined` error at the declaration site and will skip checking its usages. This effect has caused the No Preamble variant in Table 4 to have a lower error count than other two other variants since it tends to predict a lot more non-existing types. But we have verified that it was not the cause of the type error count improvement by our two-pass iterative decoding model.

## A.9    REAL EXAMPLES PRODUCED BY TYPET5

We run TypeT5 on the actual code corresponding to the example shown in Figure 1 and show the obtained preamble (Figure 5), usee context (Figure 6), main code (Figure 7), and user context (Figure 8) for the `ModelWrapper.predict` element. For each type predicted by the model, we indicate whether it is correct with a green or red marker, and if not, also show the ground truth type in red. Note that these outputs were generated by the model in the second pass of the iterative decoding process (described in subsection 3.4), so all the elements have already been annotated at least once (but some may have not been annotate the second time).

```
🐍 model_input.py                                              Preamble
 1      import random
 2      from copy import copy, deepcopy
 3      import numpy as np
 4      from collections import Counter
 5      from mypy_extensions import mypyc_attr
 6      from torch import Tensor
 7      from transformers.data.data_collator import DataCollatorForSeq2Seq
 8      from typing import NamedTuple, overload
 9      from datasets.arrow_dataset import Dataset
10      from torch.utils.data import DataLoader, RandomSampler
11      from.data import (
12          ChunkedDataset,
13          CtxArgs,
14          TokenizedSrcSet,
15          output_ids_as_types,
16          preds_to_accuracies,
17      )
18      from.type_env import AccuracyMetric, PythonType
19      from.utils import *
20      @dataclass
21      class DecodingArgs:
22          ...
23      @dataclass
24      class DatasetPredResult(Generic[T1]):
25          ...
26      @dataclass
27      class ModelWrapper:
28          ...
```

Figure 5: The preamble gathers all the important statements and class headers from the current file. This helps the model see which types are available and where each symbol comes from.

```
71     # spot.model                                              Usee Context
72     @dataclass
73     class DecodingArgs:
74         sampling_max_tokens: int ✓
75         ctx_args: CtxArgs ✓
76
77     # spot.utils
78     def assert_eq(x: T1, *xs: T1, extra_message: Callable[[], str] = lam
79                       ✓              ✓                         ✓
80     # spot.model
81     @dataclass
82     class DatasetPredResult(Generic[T1]):
83         chunks: ChunkedDataset ✓
84         predictions: list[T1] ✗ list[list[PythonType]]
85         extra_info: list[dict] = field(default_factory=list)
86                        ✗ list[T1]
87     # spot.model
88     @dataclass
89     class ModelWrapper:
90         model: PythonType ✗ ModelSPOT
91         tokenizer: PythonType ✗ TokenizerSPOT
92         args: CtxArgs ✗ DecodingArgs
93         def predict_on_batch(
94             self,
95             batch: dict, ✓
96             num_return_sequences: int = None, ✓
97         ) -> tuple[list[PythonType], Tensor]:...
98                ✗ tuple[list[list[PythonType]], Tensor]
99
100    # spot.model
101    def dynamic_dataloader(
102        dataset: ChunkedDataset, ✗ Dataset
103        max_tokens: int, ✓
104        collate_fn: DataCollatorForSeq2Seq, ✓
105        shuffle: bool = False, ✓
106    ) -> DataLoader:...
107
108    # Used above
```

Figure 6: The usee context shows the signature of the elements that are used by the main code or by elements from the user context. By seeing their predicted type signatures, the model can understand the type-level behavior of these definitions without having to dive into their implementation.

```
109      # spot.model                                                    Main Code
110      @dataclass
111      class ModelWrapper:
112          def predict(
113              self,
114              dataset: ChunkedDataset, ❌ Dataset
115              tqdm_args: dict = {}, ✅
116              num_return_sequences: int = None, ✅
117          ) -> list[PythonType]: ❌ list[list[PythonType]]
118              model = self.model
119              collator = DataCollatorForSeq2Seq(self.tokenizer, model)
120              loader = dynamic_dataloader(
121                  dataset,
122                  max_tokens=self.args.sampling_max_tokens,
123                  collate_fn=collator,
124                  shuffle=True,
125              )
126              device = model.device
127              pred_types = dict[int, list]()
128              with tqdm(
129                  total=len(dataset), desc="predict", smoothing=0.01, **tqdm_args
130              ) as tqdm_bar:
131                  for batch in loader:
132                      n_chunks = batch["input_ids"].shape[0]
133                      batch["input_ids"] = batch["input_ids"].to(device)
134                      preds, _ = self.predict_on_batch(batch, num_return_sequences)
135                      for i, c_id in enumerate(batch["chunk_id"]):
136                          c_id = int(c_id)
137                          if num_return_sequences is None:
138                              pred_types[c_id] = preds[i]
139                          else:
140                              pred_types[c_id] = preds[
141                                  i * num_return_sequences : (i + 1) * num_return_seque
142                              ]
143                      tqdm_bar.update(n_chunks)
144              return [pred_types[int(c_id)] for c_id in dataset["chunk_id"]]
145
```

Figure 7: The main code is the element that is being annotated by the model at the current decoding step. The model has made two errors in this example, both of which can be directly attributed to the previous two errors made in the usee context (line 102 and line 97). This shows that the model is making coherent predictions according to the context, and such errors can be avoided if the user has corrected the previous errors (as described in subsection 4.4).

```
147     # Users below                                    User Context, first half
148     # spot.model
149     @dataclass
150     class ModelWrapper:
151         def eval_on_dataset(
152             self,
153             src_data: TokenizedSrcSet, ✅
154             max_labels: int = None, ✅
155             tqdm_args: dict = {}, ✅
156         ) -> DatasetPredResult: ✅
157             ctx_args = self.args.ctx_args
158             if max_labels is not None:
159                 ctx_args = copy(ctx_args)
160                 ctx_args.max_labels = max_labels
161
162             chunks = src_data.to_chunks(ctx_args, tqdm_args=tqdm_args)
163             preds = self.predict(
164                 chunks.data, num_return_sequences=None, tqdm_args=tqdm_args
165             )
166             return DatasetPredResult(chunks, preds)
167
```

```
169     # spot.decode                                   User Context, second half
170     def sample_candidates(
171         wrapper: ModelWrapper, ✅
172         src_data: TokenizedSrcSet, ✅
173         n_samples: int, ✅          tuple[ChunkedDataset, list[list[list[PythonType]]]]
174     ) -> tuple[ChunkedDataset, list[PythonType]]: ❌
175         ctx_args = wrapper.args.ctx_args
176         do_sample = wrapper.args.do_sample
177         if not do_sample:
178             assert wrapper.args.num_beams is not None, "num_beams needs to be set"
179             assert n_samples <= wrapper.args.num_beams
180
181         chunks = src_data.to_chunks(ctx_args)
182         n_chunks = len(chunks.data)
183
184         if do_sample:
185             samples = [
186                 wrapper.predict(chunks.data, tqdm_args={})
187                 for _ in tqdm(range(n_samples), desc="Sampling")
188             ]
189  >      else: ⋯
197
198         def get_preds(chunk_id, sample_id):
199             return (
200                 samples[sample_id][chunk_id] if do_sample else samples[chunk_id][sampl
201             )
202
203         pred_candidates = [
204             [get_preds(cid, sid) for sid in range(n_samples)] for cid in range(n_chunk
205         ]
206         return chunks, pred_candidates
207
```

Figure 8: The user context shows two callers of the `predict` method from the main code. We see that the model successfully predicts all user-defined types, despite the fact that these are all new classes defined in the current project.

