# OpenReview forum: "TypeT5: Seq2seq Type Inference using Static Analysis"
_ICLR.cc/2023/Conference — ICLR 2023 poster_

### Official Review · Reviewer_JQ8L · 2022-10-24

**Confidence:** 5
**Correctness:** 3
**Technical Novelty And Significance:** 3
**Empirical Novelty And Significance:** Not applicable
**Recommendation:** 6

**Clarity, Quality, Novelty And Reproducibility:**

Overall, the paper is quite well written and reasonably novel. I list a few typos below. The data processing steps and model implementation details are reasonably well documented, including in helpful appendices.

Typos:
- P1: missing space between "sythesis" and "(Li". Same on previous line
- P3: extra space after "usees".
- P4: "and edge" -> "an edge"
- P8: "it allow" -> "it allows"
- P8: "On BetterTypes4Py ..." -> "On the BetterTypes4Py"


**Strength And Weaknesses:**

**Update:** The authors have provided additional results in response to my concerns below, which underscore that the TypeT5 consistently improves over CodeT5 on rare types, and provide both additional ablations under the requested conditions and information on the training and inference _cost_. I have thus increased my score.

This proposed approach balances the use of powerful neural language models with a clear understanding of how developers type-annotate their projects. The result is an intuitive and rigorous approach that yields improvements over all alternative methods. Working against it is the fact that the improvements over a plain CodeT5 model are quite small, which stands in especially stark contrast to how vastly that model in turn improved over existing baselines. This is no fault of the authors -- pretrained language models are hard to beat -- so I am inclined to not consider this a major problem, but it does imply that the paper should make a stronger argument for its current performance. Specifically, three parts of the reporting should be expanded:

1. The emphasis on (or alternatively, the definition of) adjusted accuracy feels unwarranted. While I could see filtering out `None`/`Any` labels as a reasonable step, replacing fully qualified types with simple ones and dropping `Optional` types would alter the type distribution quite significantly. This matters because the gain of TypeT5 over CodeT5 is quite a bit larger under the "adjusted" metric as it is under the "full" one, suggesting that it does better at inferring comparatively less precise annotations. Given this, please provide a second ablation for Tab. 4 focusing on full accuracy, so as to capture the factors that make TypeT5 outperform CodeT5 in the more challenging setting.

2. One of the paper's stated goals (starting from the abstract) is to yield better type predictions on rare and complex types. That it realizes the "complex" part of this felt convincing, given the results in Tab. 2 & 3. It is unclear, however, why Tab. 3 does not show the accuracy solely on _rare_ types, but rather combines them with the common type results that were already shown separately before. Given that common types are far more common than rare ones and thus contribute predominantly to these results, I assume that the accuracy drop between tables 2 & 3 signals a far lower accuracy on rare types for all models. Please provide an equivalent table with the rare type results only (and, preferably, replace Tab. 3 in the paper with said table). There is no harm at all in reporting low accuracies on this task; it is objectively hard for other models as well. But the current setup makes it very hard to gauge whether the paper achieved this sub-goal -- if anything, in some cases it looks like it failed to; e.g., on InferTypes4Py, CodeT5 drops about 9% (points) from Tab. 2 to Tab 3. (full accuracy) while TypeT5 drops 10%, suggesting that it performs *worse* on rare types than the former, or at least, that its performance improvement over CodeT5 is much slimmer on rare types. This is important information.

3. A modest performance gain should be presented along with the increase in cost, if any, that accompanies using the more performant technique relative to its baseline. Please discuss how much more expensive TypeT5 is, both for training (in terms of data collection, training time), and inference (time, memory), than CodeT5.

**Summary Of The Paper:**

This work adopts a language modeling approach to tackle program type prediction. Specifically, it fine-tunes CodeT5 on a type annotation task, and extends the standard approach by both including richer sets of context inferred via static analysis, and ordering the generation process to simulate information flow through a program usage graph. The resulting model provides reasonable improvements over a baseline approach that just uses CodeT5, which in turn performs far better than prior work.

**Summary Of The Review:**

The work is well written and provides a natural approach that balances the performance potential of a pretrained language model with a series of improvements that provide the model with access to information that is pertinent to type annotation. The resulting model performs better than CodeT5, but only by a modest margin, and the evaluation leaves several open questions about the interpretation of this improvement. My review asks for expanded results that, if in line with the narrative in the paper, should make the improvements more convincing.

---

> ### Author Response · Authors · 2022-11-10
> **Response to Reviewer 5 (JQ8L)**
>
> We thank Reviewer JQ8L for their positive comments and helpful feedback on our work. We now respond to specific comments below.
>
> ---
> > The emphasis on (or alternatively, the definition of) adjusted accuracy feels unwarranted. While I could see filtering out None / Any labels as a reasonable step, replacing fully qualified types with simple ones and dropping Optional types would alter the type distribution quite significantly. This matters because the gain of TypeT5 over CodeT5 is quite a bit larger under the "adjusted" metric as it is under the "full" one, suggesting that it does better at inferring comparatively less precise annotations. Given this, please provide a second ablation for Tab. 4 focusing on full accuracy, so as to capture the factors that make TypeT5 outperform CodeT5 in the more challenging setting.
>
> We have added a detailed version of Table 3 and 4 in the appendix to show other accuracy metrics as well.
>
> We drop qualifications before the predicted types to help us compare with prior work since they do not output context-dependent qualifications like we do (e.g., predicting `typing.List` or `List` depending on how the type is imported). Dropping top-level `Optional` is based on the observation that some projects in our dataset have configured the type checker to allow implicit `Optional` types, so it would not be possible for the model to know if it should output `Optional[T]` or `T` just from the untyped code. It is also a standard practice in other prior work (e.g., [DeepTyper](https://dl.acm.org/doi/abs/10.1145/3236024.3236051) and [LambdaNet](https://openreview.net/forum?id=Hkx6hANtwH)).
>
> ---
> > One of the paper's stated goals (starting from the abstract) is to yield better type predictions on rare and complex types. [...] Please provide an equivalent table with the rare type results only (and, preferably, replace Tab. 3 in the paper with said table).
>
> We followed the suggestion and now show rare type accuracies in Table 3. We see that TypeT5 also significantly improved upon CodeT5 on rare and complex types.
>
> ---
> > A modest performance gain should be presented along with the increase in cost, if any, that accompanies using the more performant technique relative to its baseline. Please discuss how much more expensive TypeT5 is, both for training (in terms of data collection, training time), and inference (time, memory), than CodeT5.
>
> Both CodeT5 and TypeT5 are quite fast to train: On a single Quadro RTX 8000 GPU with 48GB memory, CodeT5 takes about 3.7 hours, and TypeT5 takes about 10.6 hours. As for the inference time, on the BetterTypes4Py test set, CodeT5 takes about half an hour, and TypeT5 takes about 4 hours. CodeT5 has a much faster inference time since it benefits from batching (whereas our approach needs to perform decoding element-by-element).
>
> As for data collection, since the involved static analysis operations are fairly lightweight and we parallelize the construction of different usage graphs on CPUs, the time spent on constructing the usage graphs only makes up a tiny fraction of the total training or inference time. For example, using 28 Python processes, it only takes about 8 minutes to process the entire BetterTypes4Py training set, including the time spent on other tasks such as parsing, code transformation, and tokenization.
>
> We added these time statistics to section A.1 and A.4 in our revision.

---

> > ### Comment · Reviewer_JQ8L · 2022-11-10
> > **Re: Paper Updates**
> >
> > Thanks for your revisions and clarifications! The paper is significantly stronger now; I will revise my score to an accept.
> >
> > It's good to see that the rare types results are still clearly in favor of the proposed technique. In relation to the computational cost results: given that TypeT5 is quite a bit slower than CodeT5, I encourage the authors to mention this significant disparity in training and inference time in the main body of the text. This will make it easier for practicioners to decide whether using TypeT5 over CodeT5 based on their performance needs.

---

### Official Review · Reviewer_8rav · 2022-10-25

**Confidence:** 5
**Correctness:** 3
**Technical Novelty And Significance:** 3
**Empirical Novelty And Significance:** 3
**Recommendation:** 8

**Clarity, Quality, Novelty And Reproducibility:**

Quality of work is good, the paper was well written and the evaluation was transparent. Given the model weights and the dataset, the paper would be reproducible.

The originally of the work was fine as it is moving the needle on SOTA type inference using the latest and greatest model architectures. The application of static analysis was a good touch.

**Strength And Weaknesses:**

Strengths
- Design: the design of the model is important in the field of type inference. The application of CodeT5 was overdue and the implementation of static analysis was fitting for the model architecture
- Comparison with SOTA: Comparison with state of the art and corresponding metrics in previous works was ok.
- Use of type error count is good.
- Examples in Appendix were good.
- Ablation study was appropriately done
- Application of Human in the loop was well done

Weaknesses
- Comparisons on previous datasets in full and BetterTypes4Py/InferTypes4Py would strengthen results. The release of these datasets with the paper submission would help reviewers understand the distribution and demographic of types that are being evaluated. Table 1 is good, but the dataset would be best.

- Authors do not evaluate Typilus model on types that are not common. Authors claim that released Typilus model is not capable of predicting types outside of common (top100). The authors should verify this with Allamanis etal as it is a SOTA across rare types.

- The authors should validate claim that Type4Py inference is lower because of leakage. But in the appendix, the authors state that the Type4Py discrepancy between train and test in the figure on their dataset is due to (1) Type4Py evaluation on local variables (not just APIs) and (2) compiler inferable types (easier type subset). Both of these reasons would contribute to a higher evaluation on the original ManyTypes4Py dataset but does not explain the discrepancy between train and test on their dataset. With the availability of Type4Py and ManyTypes4Py, validating this claim should not be difficult; the authors might find that their intuition led to a tangential discovery maybe a degraded performance on novel types not available in the KNN search, for example, and not data leakage.

**Summary Of The Paper:**

The authors present a new approach to type inference by using T5 architecture to generate types in a seq2seq fashion. Their proposed model, TypeT5 is CodeT5 fine-tuned on a subset of ManyTypes4Py named BetterTypes4Py. TypeT5 uses static analysis to compose its contextual input with a preamble, main code, users, and usees. The authors found that such static analysis can improve TypeT5's type generation across a series of metrics and across the type error count. The authors also wanted TypeT5 to generate consistent type information and exploit its previous type predictions (as additional context at inference time), so they introduced sequential decoding where passes of inference are made from usee-to-user and vice versa. The model is trained normally (like T5) with teacher-forcing to directly optimizing on ground truth labels rather than its own predictions.

The results show that pre-training (CodeT5 and TypeT5) improves accuracy measures across all, simple, and complex types. The authors demonstrate that their techniques for introducing more context was helpful for the model at inference time. The static analysis required for the context appears to be lightweight. The authors introduce a type error count metric that was not previously introduced as a metric but is practically useful; consistency is equally important to accuracy and type accuracy does not indicate type consistency, thus, such a metric is helpful.

The authors found some interesting attributes to their model, namely, the importance of preamble information (which could improve other pre-trained approaches that do not use preamble information), and that user-to-usee decoding performed worse than independent predictions.

Finally, the authors took advantage of TypeT5s conditioning on previous predictions as an opportunity to correct incorrect predictions. In other words, during the sequential decoding process, if the decoding was incorrect, an human in the loop could correct the context and allow the model to continue inference with a higher precision rate.

**Summary Of The Review:**

My recommendation of this paper is based on a strong understanding of this works position in neural type inference. This paper does contribute to the existing body of research of neural type inference.

---

> ### Author Response · Authors · 2022-11-10
> **Response to Reviewer 4 (8rav)**
>
> We thank Reviewer 8rav for their positive comments and helpful feedback on our work. We now respond to specific comments below.
>
> ---
> > Comparisons on previous datasets in full and BetterTypes4Py/InferTypes4Py would strengthen results. The release of these datasets with the paper submission would help reviewers understand the distribution and demographic of types that are being evaluated. Table 1 is good, but the dataset would be best.
>
> Type4Py's test set is split by files, so the files from the same project might be split into both training and test set. Since our approach works at the project level, and also because the released dataset is restricted to their tokenization and labels, there is no easy way for us to evaluate on their dataset.
>
> We will publish our dataset, code, and the trained model weights upon paper acceptance. Our dataset is too large to be uploaded as a supplementary material.
>
> ---
> > Authors do not evaluate Typilus model on types that are not common. Authors claim that released Typilus model is not capable of predicting types outside of common (top100). The authors should verify this with Allamanis etal as it is a SOTA across rare types.
>
> We contacted the authors about this, and they confirmed that the model weights of the original model have been lost, so the only option is to retrain the model. Unfortunately, retraining Typilus appears to be difficult due to its use of outdated Python packages (in fact, we had to manually fix a few issues in order to just run the released version).
>
>
> ---
> > The authors should validate claim that Type4Py inference is lower because of leakage. But in the appendix, the authors state that the Type4Py discrepancy between train and test in the figure on their dataset is due to (1) Type4Py evaluation on local variables (not just APIs) and (2) compiler inferable types (easier type subset). Both of these reasons would contribute to a higher evaluation on the original ManyTypes4Py dataset but does not explain the discrepancy between train and test on their dataset. With the availability of Type4Py and ManyTypes4Py, validating this claim should not be difficult; the authors might find that their intuition led to a tangential discovery maybe a degraded performance on novel types not available in the KNN search, for example, and not data leakage.
>
> We do not quite understand this point. In our paper, we didn’t present results on any training set; all results are from the test sets.
>
> For the difference between Type4Py’s performance on (the test set of) ManyTypes4Py and InferTypes4Py, it’s likely because the projects in InferTypes4Py tend to use more complex types (average type size = 1.72) than ManyTypes4Py (average type size = 1.43), as shown in Table 1.

---

> > ### Comment · Reviewer_8rav · 2022-11-11
> > **Response to Authors**
> >
> > Thank you for your follow up and comments.
> >
> > - I agree with your point, that retraining TypeT5 to compare with Type4Py on the file level would not be trivial, moreover, your approach is specifically designed to operate on the project level. This is a common use case especially in evaluation (project level vs. file level).
> >
> > - I am satisfied with your answers and I will increase my score to accept.

---

### Official Review · Reviewer_2PXr · 2022-10-26

**Confidence:** 4
**Correctness:** 4
**Technical Novelty And Significance:** 3
**Empirical Novelty And Significance:** 3
**Recommendation:** 6

**Clarity, Quality, Novelty And Reproducibility:**

The paper is clear and easy to follow. The most significant contribution to achieving the reported performance is using CodeI5. However, the paper provides several inventions to improve the performance and makes type inference more coherent. The appendix also provides more information to explain the part unclear in the paper.

**Strength And Weaknesses:**

Strengths:

+ The proposed approach, especially the use of CodeT5, succeeds in inferring user-defined and parametric types.

+ The paper experimentally demonstrated that contextual information and the sequential decoding improve accuracy and make type inference more coherent.

+ The documentation is well written.

Weaknesses:

- The paper could provide a finer-grained analysis of the experimental result. Specifically, I am curious about the model's performance for parametric and under-defined types because this is the first work that addresses both kinds of types. Unfortunately, I cannot find it in the paper because all the accuracy numbers in Tables 2, 3, 4, and 5 are for the entire dataset, which includes primitive types such as int.

- The paper does not report the cost of constructing user graphs. Because the construction needs to examine the overall codebase and the libraries it uses in the worst case, it might take too much time. Perhaps in practice the graph construction is not problematic, but I'm unsure whether it is the case.

- (Minor) The notion of coherence needs to be better explained in the body of the paper. The footnote seems far from the formal definition of coherence provided in A.6.

**Summary Of The Paper:**

With the aim of type inference for untyped code, the paper feeds, in addition to tokenized source code, contextual information gained from static analysis to the seq2seq-based code completion model called CodeT5. CodeT5 itself can infer types of elements in untyped code (including user-defined and parametric types) successfully, but incorporating the contextual information improves the accuracy performance marginally. Furthermore, the paper provided the idea of sequential decoding, which refines the contextual information gradually using the inference result of the model, and showed that the use of sequential decoding makes type inference more coherent.

**Summary Of The Review:**

The paper used CodeT5 to infer types and provided methods to improve the performance of the model with respect to accuracy and coherence. It would be nicer to provide a finer-grained analysis of the experimental result and the cost of constructing user graphs.

---

> ### Author Response · Authors · 2022-11-10
> **Response to Reviewer 3 (2PXr)**
>
> We thank Reviewer 2PXr for their positive comments and helpful feedback on our work. We now respond to specific comments below.
>
> ---
> > The paper could provide a finer-grained analysis of the experimental result. Specifically, I am curious about the model's performance for parametric and under-defined types because this is the first work that addresses both kinds of types. Unfortunately, I cannot find it in the paper because all the accuracy numbers in Tables 2, 3, 4, and 5 are for the entire dataset, which includes primitive types such as int.
>
> We added a new Table 3 to show the performance on rare types. We also included more detailed accuracies for Table 4 and 5 in the appendix. From the new Table 3 we can see that TypeT5 also significantly improved upon CodeT5 on rare and complex types.
>
> ---
> > The paper does not report the cost of constructing user graphs. Because the construction needs to examine the overall codebase and the libraries it uses in the worst case, it might take too much time. Perhaps in practice the graph construction is not problematic, but I'm unsure whether it is the case.
>
> In this work, we only construct the usage graph for things defined in the current project. Since the involved static analysis operations are fairly lightweight and we parallelize the construction of different graphs on CPUs, the time spent on constructing the usage graphs only makes up a tiny fraction of the total training or inference time. For example, using 28 Python processes, it only takes about 8 minutes to process the entire BetterTypes4Py training set, including the time spent on other tasks such as parsing, code transformation, and tokenization. We added these time statistics in the appendix in our revision.

---

### Official Review · Reviewer_SGAP · 2022-11-01

**Confidence:** 4
**Correctness:** 4
**Technical Novelty And Significance:** 3
**Empirical Novelty And Significance:** Not applicable
**Recommendation:** 6

**Clarity, Quality, Novelty And Reproducibility:**

## Feedback for the authors
### Major comments

- I would appreciate seeing a comparison between TypeT5 and HiTyper [2] in the evaluation section. HiTyper [2] is based on the combination of static analysis and deep learning. It also outperforms Type4Py in the prediction of rare and user-defined types. Also, HiTyper should have been mentioned in related work.
- For the evaluation, developer-provided type annotations are used, which are not always sound or coherent [3]. This might be a potential threat to the validity of the obtained results. To address this, in Type4Py’s paper, the authors used a type-checked dataset for both training and evaluation. Also, using a type-checked dataset might change the produced type errors by TypeT5. Hence, I highly suggest to type-check ground truth in the dataset using a type checker, e.g., Mypy.
- As mentioned in the text, Type4Py’s performance is pretty low compared to its original paper. I believe this is because Type4Py was trained on a different dataset and also different type normalization rules were used for BetterTypes4Py. This is also true for Typilus. I assume BT4Py’s projects were cloned in 2022 whereas MT4Py’s projects were gathered in Sep. 2020. For a fair comparison, I highly recommended to re-train both Type4Py and Typilus on BT4Py and evaluating them on the test set of BT4Py and IT4Py.
- I highly suggest including Typilus in Table 3, “Accuracy comparison on all types (common + rare).” to have a consistent and rigorous comparison in the evaluation section.
- It should be pointed out that in both Type4Py and Typilus papers, the depth of the parametric types is reduced to 2. For example, the type annotation List[List[Tuple[int]]] is converted to List[List[Any]]]. This conversion is not performed for the BetterTypes4Py dataset, which makes it even harder for Type4Py and Typilus to predict some complex types with deep nested levels. This is another reason why I believe that both Type4Py and Typilus should be re-trained on BT4Py with the same type normalization rules.

### Minor comments
- In the ablation study, TypeT5 produces quite a number of type errors, i.e., ~5K. Given this, a critical reader might question the usefulness and practicality of TypeT5 if used by developers. I would suggest showing a percentage of type errors considering all the predictions made for code elements in the test set, similar to what was done in Typilus’ paper regarding type-checking the model’s predictions.
- Looking at Table 5, the two-pass decoding scheme produces negligible accuracy improvement over the other strategies (e.g., Random). One might ask is it worth using considering its additional overhead at inference time? Also, with TwoPass, the number of type errors still seems high. Based on the reported results in Table 5, the two-pass decoding scheme does not seem convincing to use. Maybe adding inference time to Table 5 might better justify using the TwoPass strategy.
- In the text, it is unclear what are exactly the code elements used to construct the usage graph? Only two examples were provided in the text for functions and variables. Please clarify this in the text and, if needed, provide a comprehensive list of code elements in the Appendix, which are extracted to build a usage graph. This helps to reproduce the results of the paper.

## Questions for the authors
- Is the BetterTypes4Py dataset de-duplicated? Code duplication [1] has been shown to have adverse effects on the performance of ML models and it would blur the results. It is essential to perform this step.
- What dataset is exactly used in subsection 4.3 for the ablation study of TypeT5? I could not find it in the text. Please clarify this in the text. I assume it is BetterTypes4Py by comparing the accuracy of TypeT5 in Tables 3 and 4.


## References
[1] Allamanis, M. (2019, October). The adverse effects of code duplication in machine learning models of code. In Proceedings of the 2019 ACM SIGPLAN International Symposium on New Ideas, New Paradigms, and Reflections on Programming and Software (pp. 143-153).

[2] Peng, Y., Gao, C., Li, Z., Gao, B., Lo, D., Zhang, Q., & Lyu, M. (2022, May). Static inference meets deep learning: a hybrid type inference approach for python. In Proceedings of the 44th International Conference on Software Engineering (pp. 2019-2030).

[3] Ore, John-Paul, et al. "Assessing the type annotation burden." Proceedings of the 33rd ACM/IEEE International Conference on Automated Software Engineering. 2018.



**Strength And Weaknesses:**

### Pros
- A novel type inference technique based on CodeT5, which leverages global information in code and performs two-pass sequential decoding for inference.
- The paper is well-written and has a good presentation. The proposed approach is explained with sufficient detail to reproduce the work.

### Cons
- The evaluation of the proposed approach is not fully sound and rigorous. The evaluation has some (major) shortcomings that need to be addressed. I have explained the shortcomings in “feedback for the authors”.


**Summary Of The Paper:**

Type inference is a challenging task for the Python language given its dynamic nature. Scholars have recently proposed machine learning (ML)-based techniques to infer types for Python. Previous techniques infer types based on seen examples during training, i.e, they cannot synthesize types, which hinders their ability to infer project-specific types. This paper presents TypeT5, a CodeT5-based model, whose main idea is to create a usage graph to capture usee-user relationships with static analysis, which provides global information to the CodeT5 model about code elements that need to be typed. Also, the proposed model has a two-pass sequential decoding that is conditioned based on previous type predictions. This allows bidirectional information flow from usees to users given a usage graph. Overall, the experimental results show that TypeT5 outperforms the state-of-the-art type inference models, namely, Type4Py and Typilus.

**Summary Of The Review:**

This paper presents a novel technique, TypeT5, to tackle the type inference challenge for Python. Also, treating the type inference task as code completion is new. Overall, I believe that this paper has good potential to become an influential work in ML-based type inference research. However, I have some (major) comments regarding the soundness of the evaluation of TypeT5, which hampers the paper from reaching its full potential and impact. I briefly mention the main comments here as they are explained in detail later in the review.

- Human-provided type annotations are used as ground truth, which is not always valid [3]. This can harm the soundness of the obtained results in the evaluation.
- The state-of-the-art approaches, i.e., Type4Py and Typilus, are not fairly represented and evaluated.
- Lack of comparison with a very similar published work, HiTyper [2], which is also a very recent SOTA type inference model for Python.

To increase the recommendation score to the acceptance level, all three raised concerns regarding the evaluation should be addressed.

---

> ### Author Response · Authors · 2022-11-10
> **Response to Reviewer 2 (SGAP)**
>
> We thank Reviewer SGAP for their positive comments and helpful feedback on our work. We now respond to specific comments below.
>
> ---
> > I would appreciate seeing a comparison between TypeT5 and HiTyper [2] in the evaluation section. HiTyper [2] is based on the combination of static analysis and deep learning. It also outperforms Type4Py in the prediction of rare and user-defined types. Also, HiTyper should have been mentioned in related work.
>
> As we mentioned in “Major Updates” above,  we included new discussion and results in our new revision. We note that the core HiTyper algorithm is agnostic to the ML backend; in some sense, we can view it as a sophisticated decoding strategy that works with any ML model, including ours. Furthermore, the set of type inference rules HiTyper implemented does not cover many real-world Python features such as generic user-defined types, structural types (protocol types), or the deliberate use of the Any type (which enables gradual typing but cannot be type-checked). They also apply a very simple name-based algorithm to predict user-defined types. As a result, we see the performance of HiTyper is still much lower than our model’s.
>
> ---
> > For the evaluation, developer-provided type annotations are used, which are not always sound or coherent [3]. This might be a potential threat to the validity of the obtained results. To address this, in Type4Py’s paper, the authors used a type-checked dataset for both training and evaluation. Also, using a type-checked dataset might change the produced type errors by TypeT5. Hence, I highly suggest to type-check ground truth in the dataset using a type checker, e.g., Mypy.
>
> We acknowledge that the human type annotation quality varies in our dataset. This partially motivated us to construct the InferTypes4Py dataset, which consists of high-quality type annotations with a very low error rate. In particular, our own codebase (which is part of InferTypes4Py) makes heavy use of type annotations throughout the development process and is continuously type-checked by VSCode.
>
> However, passing the type checker does not guarantee correctness, and it is well known that type errors can manifest far away from their true source (e.g., [Pavlinovic et al](https://arxiv.org/abs/1508.06836)). The proposed method in Type4Py simply removes one type annotation at a time until no type errors are left in the current file. Since there are generally more correct types than incorrect ones, we found that this procedure tends to remove correct types rather than the incorrect ones. As a contrived example to illustrate this, in the following code snippet, both `make` and `use` has an incorrectly annotated parameter and trigger two type errors, but the proposed procedure will start by removing `Foo.bar`’s correct type annotation, which will make both errors go away and keep the incorrect annotations intact.
>
> ```python
> class Foo:
> 	bar: str  # bar is correct
>
> def make(x: int): # x should be str
> 	return Foo(x)  # error: wrong arg type
>
> def use(foo, y: int): # y should be str
> 	return foo.bar + y  # error: cannot add str and int
> ```

---

> > ### Author Response · Authors · 2022-11-10
> > **Response to Reviewer 2, Continued**
> >
> > (Moved this response to outer scope.)

---

> ### Author Response · Authors · 2022-11-10
> **Response to Reviewer 2, Continued**
>
> > As mentioned in the text, Type4Py’s performance is pretty low compared to its original paper. I believe this is because Type4Py was trained on a different dataset and also different type normalization rules were used for BetterTypes4Py. This is also true for Typilus. I assume BT4Py’s projects were cloned in 2022 whereas MT4Py’s projects were gathered in Sep. 2020. For a fair comparison, I highly recommended to re-train both Type4Py and Typilus on BT4Py and evaluating them on the test set of BT4Py and IT4Py.
>
> We didn’t apply the type normalization rules during training time in any way but only used them to compute the accuracies. We try to make our type normalization rules similar to the two prior papers when possible. Particularly, the base accuracy we report is almost identical to the two papers’ original definition, so we believe the large performance gaps on base accuracy (e.g. ours 84.82 vs 47.51) cannot be explained by metric differences.
>
> It’s true that our dataset may lead to a different label distribution; however, retraining all models also does not make the comparison absolutely fair since we are using a fine-tuning approach. For example, since our BT4Py dataset uses a subset of the repos from MT4Py, training Type4Py on it will significantly reduce the amount of training data available as well as the set of types in its prediction space (since it only predicts types from its training set), which is likely to further hurt its performance. Our fine-tuning approach is less data-hungry since we benefit from CodeT5’s large-scale, multilingual pre-training (which uses more than 8 million functions from 8 programming languages), but it would be infeasible to train Type4Py and Typilus on these data, so we believe reusing the published model weights is a fairer setting (as was done in [TypeBert](https://dl.acm.org/doi/abs/10.1145/3468264.3473135)).
>
> As we mentioned in “Major Updates”, after manual inspection, we believe our results correctly represent Type4Py’s real-world performance. We encourage the reviewer to take a look at the uploaded model predictions to verify this claim.
>
> ---
> > I highly suggest including Typilus in Table 3, “Accuracy comparison on all types (common + rare).” to have a consistent and rigorous comparison in the evaluation section.
>
> The published version is only able to predict common types. We have confirmed this with the author. Unfortunately, retraining Typilus appears to be difficult due to its use of outdated python packages (in fact, we had to manually fix a few issues in order to just run the released version).
>
> > It should be pointed out that in both Type4Py and Typilus papers, the depth of the parametric types is reduced to 2. [...] This conversion is not performed for the BetterTypes4Py dataset, which makes it even harder for Type4Py and Typilus to predict some complex types with deep nested levels.
>
> We believe the two prior approaches set this depth limit not because of validity but because of their own limitations. Since they treat each type as an independent entity whose embedding vector needs to be trained from scratch, increasing the depth limit can lead to an explosion of types, which will likely further hurt their performance.
>
> To address your concern, we instead modified our accuracy metric to follow the same depth limit during normalization and remeasured Type4Py and Typilus’ performance on BetterTypes4Py, shown below:
>
>
> __Type4Py Performance on all types__
> | Type Depth | all   | simple | complex |
> |------------|-------|--------|---------|
> | Unlimited  | 34.52 | 35.87  | 19.68   |
> | limit=2    | 34.55 | 35.87  | 19.94   |
>
> __Typilus Performance on common types__
> | Type Depth | all   | simple | complex |
> |------------|-------|--------|---------|
> | Unlimited  | 54.05 | 55.12  | 33.23   |
> | limit=2    | 54.05 | 55.12 | 33.23  |
>
> As we can see, changing this depth limit has almost no impact on the overall accuracy (Typilus's performance change was less than 0.01% since there are very few complex common types).
>
> ---
> > In the ablation study, TypeT5 produces quite a number of type errors, i.e., ~5K. Given this, a critical reader might question the usefulness and practicality of TypeT5 if used by developers. I would suggest showing a percentage of type errors considering all the predictions made for code elements in the test set, similar to what was done in Typilus’ paper regarding type-checking the model’s predictions.
>
> We have a total of 28.4K types to predict in the test set, so this roughly translates to one error every 5.68 types.
> Since the lack of type errors does not guarantee correctness with respect to user intention, we believe the user-guided decoding scheme proposed in section 4.4 is the most practical way to ensure that the resulting types are both correct and type-check.

---

> ### Author Response · Authors · 2022-11-10
> **Response to Reviewer 2, Continued**
>
> > Looking at Table 5, the two-pass decoding scheme produces negligible accuracy improvement over the other strategies (e.g., Random). One might ask is it worth using considering its additional overhead at inference time? Also, with TwoPass, the number of type errors still seems high. Based on the reported results in Table 5, the two-pass decoding scheme does not seem convincing to use. Maybe adding inference time to Table 5 might better justify using the TwoPass strategy.
>
> TwoPass simply doubles the decoding cost of single-pass strategies. Our BT4Py test set contains a total of 16881 code elements, and TwoPass takes 3.9 hours to perform inference, resulting in an average speed of 1.2 elements / second.
> We added more time statistics in Section A.4 in our revision.
>
> ---
> > In the text, it is unclear what are exactly the code elements used to construct the usage graph? Only two examples were provided in the text for functions and variables. Please clarify this in the text and, if needed, provide a comprehensive list of code elements in the Appendix, which are extracted to build a usage graph. This helps to reproduce the results of the paper.
>
> A code element is either a top-level function (methods and global functions) or top-level variable (attributes or global variable). We describe how we encode code elements in section 3.3 under the “main code” paragraph, but we will clarify this more in our revision.
>
> ---
> > Is the BetterTypes4Py dataset de-duplicated? Code duplication [1] has been shown to have adverse effects on the performance of ML models and it would blur the results. It is essential to perform this step.
>
> It’s hard to do a file-level deduplication in our project-based setting since we need all files to be present during inference. Our InferTypes4Py dataset partially addresses this issue since we have manually verified that it does not contain files that are copy-pasted from elsewhere.
>
> We also run the popular code duplication tool [jscpd](https://github.com/kucherenko/jscpd) on our test set to detect duplicated code blocks. The analysis shows that there is relatively little duplication in the dataset (~4% of duplicated lines), and the majority of these duplications came from the same project rather than across projects, so we believe code duplication is not a major issue under our by-project evaluation.
>
> > What dataset is exactly used in subsection 4.3 for the ablation study of TypeT5? I could not find it in the text. Please clarify this in the text. I assume it is BetterTypes4Py by comparing the accuracy of TypeT5 in Tables 3 and 4.
>
> Yes, we use BetterTypes4Py in this section. We modified the table captions to emphasize this in our revision.

---

### Official Review · Reviewer_7ekP · 2022-11-04

**Confidence:** 4
**Correctness:** 3
**Technical Novelty And Significance:** 3
**Empirical Novelty And Significance:** 4
**Recommendation:** 6

**Clarity, Quality, Novelty And Reproducibility:**

**Clarity/Quality**

Overall the paper is very well written, however I have a few comments/questions:

* In Figure 4, why is the line from `eval_on_dataset` to `predict` dotted? Isn't there an explicit call (eval.py, line 12)?

* Page 5 under "Usees" : I'm a little unclear about why $\textbf{s}_\text{usee}$ is constructed like this. If a function `foo` calls two functions `bar` and `abc`, then `foo` is in users(`bar`), and `abc` is in usees(`foo`). So $\textbf{s}_\text{usee}(\texttt{bar})$ would contain `abc` also? Why would that be beneficial?

* In the two-pass system (forward and backward), do you retain type predictions from the forward pass while doing the backward pass? What if they contradict?

* In Table 2, 4 and 5, it would be nice if the best model in each column could be made bold.

**Novelty**

This approach is conceptually simple - static analysis, encoding, and a seq2seq model, along with some miscellaneous details (2 pass, etc). However, the performance gains are very impressive, and the evaluation is thorough. Therefore, this has considerable *empirical* novelty.

**Reproducibility**

As far as I can see, the tool has not yet been made publicly available. The description of the approach is reasonably thorough, and although there are a few unclear details in the approach, I think it should be possible to replicate (possibly with some assistance from the authors).

**Strength And Weaknesses:**

**Strengths**

1. Very precisely written. Excellent paper structuring that makes it easy to understand the approach. The motivating example is insightful and clearly illustrates both the problem and the benefits of the proposed solution.

1. I appreciate the thorough dataset setup as described in 4.1. I think it's very important to ensure that train and test data are properly separated. Interesting to see that Type4Py's performance falls by so much.

1. Very large performance increases. 55% to 80% is quite impressive.

**Weaknesses**

* Why do you evaluate only on public APIs, especially when you mention that that is a major reason for the drop in performance of Type4Py (Appendix A.5)?

* I agree with the point about not including "simple" labels inferable from Pyre. But just to be thorough, it would be nice to include the results for the exact same setting as Type4Py. It would be okay even if Type4Py performed better than TypeT5 in this setting (no free lunch - every model has an inductive bias, but not all biases are bad!)

EDIT - adding one more crucial point that slipped my attention in the initial review.

* Do you re-train the baseline models on your custom (filtered) dataset? As in, when you present the results of Type4Py etc, do you retrain/fine-tune them on your collected dataset or do you just use their provided weights? It is vital to retrain. I am now wondering whether some of the extremely large performance improvement is due to these models not being trained on your dataset.

**Questions for the authors**

* Why not evaluate the coherence error (Table 4 and 5) on the baseline models Typilus and Type4Py?

* UserToUsee performs worse than independent. Why is it a "bad ordering"? As shown in your motivating example, aren't there cases where you need User info before Usee, and vice-versa?



**Summary Of The Paper:**

This paper proposes a system to predict types in untyped or partially typed code. The authors propose to treat type prediction as a code completion problem, which can be solved using transformer models trained for code. However, just applying a transformer model "out of the box" would fail because there can be caller-callee dependencies in the code, sometimes across files, and these dependencies are essential to correctly predict types.

To solve this problem, the authors use static analysis to form a usage graph of these dependencies, and encode these as strings to provide additional context for the model while making predictions. They traverse the usage graph according to a topological ordering, and make predictions for each function sequentially. When the model makes a prediction for types in a function, these are carried forward to be used while predicting types for the next function in the sequence. Thus, the model is "conditioned" upon its previous type predictions.

They train their model using the "ground truth" type predictions, and evaluate it on a dataset of Python code without any type annotations. They show that their model achieves a large improvement in prediction accuracy (from ~55% to ~80%) as compared to other state of the art models.

**Summary Of The Review:**

This paper presents an approach to automatically annotate types in Python code. The approach is conceptually simple and I have some concerns about the construction of the dataset, but a) the paper is well written and well structured, b) the evaluation is very thorough, and c) the performance gains are impressive. Given these factors, I would recommend acceptance.

I am slightly less than confident in my evaluation because this is a very active research problem (predicting missing type annotations), and I am not familiar with all the related work. It is possible that I have overlooked a very similar approach or a relevant baseline that should have been compared against.

---

> ### Author Response · Authors · 2022-11-10
> **Response to Reviewer 1 (7ekP)**
>
> We thank Reviewer 7ekP for their positive comments and helpful feedback on our work. We now respond to specific comments below.
>
> ---
> > Why do you evaluate only on public APIs, especially when you mention that that is a major reason for the drop in performance of Type4Py (Appendix A.5)?
>
> We exclude all local variables in our evaluation since their types are generally easy to predict and less useful (since predicting them has no effect outside of the local scope). In fact, these local variables can nearly always be determined from other types using rule-based type inference algorithms, as was done in [HiTyper](https://github.com/JohnnyPeng18/HiTyper). As a result, programmers rarely annotate local variables (in our dataset, only about 3% of the total human annotations are local variables); hence, even if we included these local variable annotations into our accuracy calculation, it will likely have very little impact on the final result, since we calculate accuracy based on the human annotations.
>
> We believe Type4Py’s performance discrepancy is mainly due to including Pyre-inferred types—which are mainly unannotated local variables and return types and outnumber user annotations—into their test set, making their test set labels much easier than ours on average.
>
> We updated the term from “public APIs” to the more accurate description “top-level code elements” in our revision.
>
> ---
> > I agree with the point about not including "simple" labels inferable from Pyre. But just to be thorough, it would be nice to include the results for the exact same setting as Type4Py.
>
> We didn’t include Pyre-inferred types in our evaluation since only evaluating on human labels is the standard setting followed by other prior work. (e.g., [TypeWritter](https://dl.acm.org/doi/abs/10.1145/3368089.3409715), [Typilus](https://dl.acm.org/doi/abs/10.1145/3385412.3385997), [DeepTyper](https://dl.acm.org/doi/abs/10.1145/3236024.3236051), [LambdaNet](https://openreview.net/forum?id=Hkx6hANtwH)). There is no guarantee that Pyre-inferred types are correct, especially when it derives from incorrect human labels, and it would also require modifying our architecture to also predict and track the types of all local variables.
>
> ---
> > Do you re-train the baseline models on your custom (filtered) dataset? [...] It is vital to retrain. I am now wondering whether some of the extremely large performance improvement is due to these models not being trained on your dataset.
>
> We didn’t retrain any of the prior approaches on our dataset since we believe doing so wouldn’t make the comparison fairer: Since our BT4Py dataset uses a subset of the repos from MT4Py, training Type4Py on it will significantly reduce the amount of training data available as well as the set of types in its prediction space (since it only predicts types from its training set), which is likely to further hurt its performance. Our fine-tuning approach is less data-hungry since we benefit from CodeT5’s large-scale, multilingual pre-training (which uses more than 8 million functions from 8 programming languages), but it would be infeasible to train Type4Py and Typilus on these data as well, so we believe reusing the published model weights is the best option (as was done in [TypeBert](https://dl.acm.org/doi/abs/10.1145/3468264.3473135)).
>
> As we mentioned in “Major Updates” above, after manual inspection, we believe our results correctly represent Type4Py’s real-world performance. We encourage the reviewer to take a look at the uploaded model predictions to verify this claim.
>
> ---
> > Why not evaluate the coherence error (Table 4 and 5) on the baseline models Typilus and Type4Py?
>
> We didn’t evaluate coherence for the two prior approaches since they clearly have much worse performance. We are not able to measure coherence errors for Typilus since it only predicts common types. The coherence error of Type4Py is 19451, which is about 280% more than TypeT5.
>
> ---
> > UserToUsee performs worse than independent. Why is it a "bad ordering"? As shown in your motivating example, aren't there cases where you need User info before Usee, and vice-versa?
>
> UseeToUser is more beneficial than UserToUsee because most functions tend to have more usees than users (since only those most “popular” functions have more users than usees; "the friendship paradox").
>
> While it’s true that both usee and user information benefits the model’s prediction, because the model is trained to condition on ground-truth context but only sees its own prediction during test time, mistakes made earlier can have adverse effects on later time steps. Since UserToUsee is less helpful than UseeToUser, it makes more mistakes, which turns out to hurt the overall performance in many cases. However, when always conditioning on corrected previous predictions (following the user-guided setting in Section 4.4), UserToUsee does achieve a better performance, having an adjusted accuracy of 77.50% (which is 5.82% higher than Independent).

---

> ### Author Response · Authors · 2022-11-16
> **Response to clarity questions**
>
> > In Figure 4, why is the line from eval_on_dataset to predict dotted? Isn't there an explicit call (eval.py, line 12)?
>
> It is dotted since we cannot be sure (without seeing the ground-truth type annotations) that the call expression `model.predict` is invoking `ModelWrapper.predict` rather than a `predict` method from other classes. In general, unless the method call is of the form `self.method`, we can only generate potential usages (i.e., dotted lines).
>
> ---
> > Page 5 under "Usees" : I'm a little unclear about why `s_{usee}` is constructed like this. If a function `foo` calls two functions `bar` and `abc`, then `foo` is in `users(bar)`, and `abc` is in `usees(foo)`. So `s_usee(bar)` would contain `abc` also? Why would that be beneficial?
>
> As you correctly pointed out, in your example, `abc` will be inside `bar`’s usee context. This may be necessary for the model to understand the elements from the user context. Reusing your example, if we have `foo` defined as `def foo(): return abc(bar)`, then this piece of code will appear in `bar`’s user context, but seeing this gives not much information about the type of `bar` unless we also know the type signature of `abc`. Hence, including `abc` into `bar`’s usee context is crucial here.
>
> ---
> > In the two-pass system (forward and backward), do you retain type predictions from the forward pass while doing the backward pass? What if they contradict?
>
> Yes, we do retain the type predictions from the first pass when doing the second (backward) decoding pass. However, note that as we described in section 3.4, the predicted types are only used to annotate the elements from the contexts; the main code is always untyped. So in this sense, they never contradict directly. However, it’s possible that some types in the usee context might be inconsistent with the types from the user context. In such cases, we rely on the model’s ability to handle inconsistency as some of its training data also contain inconsistent human annotations.
>
> ---
> > In Table 2, 4 and 5, it would be nice if the best model in each column could be made bold.
>
> Thanks for the suggestion. We have made this change in our revision.

---

### Author Response · Authors · 2022-11-10
**Major Updates**

Thanks to the reviewers for their thoughtful reviews and helpful feedback! We have included responses to each reviewer below; here we outline the major changes we have made to the paper that are likely of interest to all reviewers.

## Comparison to HiTyper
As suggested by reviewer SGAP, we added discussion of [HiTyper](https://github.com/JohnnyPeng18/HiTyper), a more recently published SOTA on probabilistic type inference, and performed additional experiments to compare its performance with our approach. HiTyper combines the strengths of a rule-based type inference algorithm with ML type inference models by only invoking the ML model on places where the inference algorithm gets stuck and propagates the typing constraints henceforth to deduce the types elsewhere.
We have included updated results from HiTyper in Table 2 and Table 3 and have updated the paper PDF. As we can see from the new tables, although HiTyper improves the performance of Type4Py (its underlying ML model), our approach still outperforms it by a large margin.

Click to view [Updated Table 2 and 3](https://anonymous.4open.science/r/TypeT5-TinyEval-FE77/New%20Tables.png).

Furthermore, as suggested by Reviewer JQ8L, we replaced Table 3 to only show accuracies on rare types and put the combined results in the appendix.  We also added more detailed versions of Table 4 and Table 5 in the appendix, which present other accuracy metrics in addition to adjusted accuracy.

## Additional examples for comparing with Type4Py
In addition to the above changes, we would also like to address the concerns raised by some reviewers regarding whether Type4Py’s performance was correctly presented. Note that it is not simple to retrain Type4Py in our setting: because we rely on a pre-trained model (CodeT5), we are able to train our approach on smaller amounts of data than these other models. As a result, re-training Type4Py will result in them using much less data as they are not built on pre-trained language models for code. In fact, it even reduces the number of types in its prediction space since it can only predict types from its training set. We believe comparing against the released tools is the most fair way to do this comparison.

To help the reviewers get a better sense of Type4Py’s and our approach’s real-world behavior, we uploaded 3 Python files annotated by both tools’ predictions. These 3 files are randomly selected from BetterTypes4Py’s test set such that they are self-contained (we inlined a few imported functions in one file) and contain mostly simple types. We encourage the interested reviewer to take a look at these files and verify that they did not use any features that are unfair to Type4Py or require retraining it. We list the anonymized link to each file below. They can also be found in the supplementary materials inside `TinyEval-results.zip`.

- [ActivityWatch/util.py](https://anonymous.4open.science/r/TypeT5-TinyEval-FE77/util.py)
- [bookclassics/goodreads.py](https://anonymous.4open.science/r/TypeT5-TinyEval-FE77/goodreads.py)
- [webwatcher/storage.py](https://anonymous.4open.science/r/TypeT5-TinyEval-FE77/storage.py)

In total, these 3 files contain 78 human labels (under adjusted accuracy), among which 66 are common types. TypeT5 achieves an adjusted accuracy of 89.74% (and a full accuracy of 91.01%), whereas Type4Py only achieves an adjusted accuracy of 42.31% (and full accuracy of 33.71%, although the lower full accuracy is partly due to Type4Py not producing context-dependent qualified names.)

---

### Decision · Program_Chairs · 2023-01-20

**Decision:**

Accept: poster

**Justification For Why Not Higher Score:**

While this is a useful approach, the problem is not of wide interest to the community nor are there not many learnings that can be transferred to other subdomains of machine learning.

**Justification For Why Not Lower Score:**

There is no good reason to reject this work. It's technically correct, improves the state-of-the-art, and provides some value to the learning-for-code community.

**Metareview: Summary, Strengths And Weaknesses:**

This paper presents a method for predicting types in code using a T5-based model. Additionally, using static analysis TypeT5 includes additional context that is helpful for the problem.

Strengths:
- A simple, yet powerful approach to a useful application.
- A interesting combination of static analysis and T5.

Weaknesses:
- No novelty with respect to deep learning models and methods.

Overall, this is a good paper and I suggest to accept it.

**Note From Pc:**

if the above contains the word "oral" or "spotlight" please see: "oral" presentation means -> notable-top-5% and "spotlight" means -> notable-top-25%. As stated in our emails, we are disassociating presentation type from AC recommendations